# Quantitative Protein Analysis of ZPB2, ZPB1 and ZPC in the Germinal Disc and a Non-Germinal Disc Region of the Inner Perivitelline Layer in Two Genetic Lines of Turkey Hens That Differ in Fertility

**DOI:** 10.3390/ani12131672

**Published:** 2022-06-29

**Authors:** Andrew Benson, Josh Steed, Mia Malloy, Adam J. Davis

**Affiliations:** Department of Poultry Science, University of Georgia, Athens, GA 30602, USA; jsteed.phd@chemlocknutrition.com (J.S.); mmalloy@ggc.edu (M.M.); ajdavis@uga.edu (A.J.D.)

**Keywords:** zona pellucida, fertility, breeders, quantitative Western blot

## Abstract

**Simple Summary:**

At ovulation, the avian egg is only surrounded by one coat of protein containing zona pellucida (ZP) proteins. These proteins serve as key components for binding sperm and initiating fertilization. In birds, most of the sperm bind around the germinal disc region of the ovulated egg, and the number of sperm that bind to this region is associated with improved fertility. Previous research has reported differences in mRNA expression of these ZP proteins between the germinal disc and non-germinal disc regions and between two lines of turkey hens (E-line, high fertility and F-line, low fertility). Until now, differences in the ZP protein abundance in this layer have not been directly measured. In the current study, the protein abundance of the ZP proteins was evaluated, using quantitative Western analyses, between genetic lines (E-line and F-line) and location on the egg (germinal disc and non-germinal disc). Several differences in protein abundance, and discrepancies with mRNA expression, are reported. These findings are the first to examine quantitative differences in the protein composition of this important protein layer and provide insight into potential protein markers that may be used in the poultry industry for improving fertility in breeding stock.

**Abstract:**

The avian inner perivitelline layer (IPVL), containing the zona pellucida (ZP) family of proteins, surrounds the ovulated ovum. In mammalian species, ZP proteins serve as key component(s) in binding sperm and initiating the acrosome reaction. Sperm binding at the germinal disc (GD) region of the IPVL initiates fertilization in avian species, and the amount of sperm binding at the GD reflects female fertility. The current research determined whether reported differences in mRNA expression in two genetic lines of turkey hens (E, high fertility and F, low fertility) translated to the protein level. ZPB2 in the IPVL is greater in the GD region compared with the nongerminal disc (NGD) region, as indicated by both mRNA and protein expression. However, protein expressions of ZPB1 and ZPC in the IPVL of E- and F-line turkey hens was in contrast to previously reported mRNA expression. The results indicate that the mRNA expression of ZP proteins at their site of synthesis in E- and F-line hens often does not directly correlate with the IPVL abundance of these proteins. The greater protein concentration of ZPB2 in the GD region compared with the NGD regions suggests that this protein may be critical for sperm binding at the GD region.

## 1. Introduction

The ovary of the mature turkey hen contains several large yolk-filled preovulatory follicles arranged in a hierarchy according to size and designated F_1_–F_n_. The largest follicle, which will typically ovulate within 24 h, is designated as the F_1_ follicle, and the second largest follicle, which will ovulate within 24–26 h following ovulation of the F_1_, is the F_2_ follicle, and so on with the remaining hierarchical follicles (F_1_–Fn). The developing oocyte in each follicle is first surrounded by its plasma membrane, followed in order by a secreted glycoprotein matrix, termed the inner perivitelline layer (IPVL), granulosa cell layer, basal lamina, and theca cell layers. Visually apparent on the surface of the yolk of each hierarchal follicle is a white, disc-shaped spot, referred to as the germinal disc (GD), which contains the haploid female pronucleus and cellular organelles. The granulosa cells at the GD region and nongerminal disc (NGD) region of the preovulatory follicles have been shown to differ in their morphological features [1,2], mitotic activity [3], steroidogenesis [4] and transcriptome [5].

The theca, basal lamina and granulosa cell layers remain attached to the ovary following ovulation of the F_1_ follicle. Therefore, the freshly ovulated, yolk-filled oocyte is only surrounded by its plasma membrane and IPVL as it is engulfed by the infundibulum of the oviduct. Spermatozoa in the infundibulum bind with the IPVL and undergo the acrosome reaction, which facilitates their penetration through the IPVL for subsequent contact with the plasma membrane [6]. There is a significantly greater occurrence of avian sperm penetrations in the IPVL overlying the GD region as compared with the NGD regions of the oocyte [7,8,9,10,11,12]. Based on immunohistochemistry and Western blotting techniques, five ZP proteins, ZPB1, ZPC, ZPA, ZPD and ZPB2, named according to the Spargo and Hope nomenclature system [13], have been identified as glycoprotein components of the avian IPVL [14,15,16,17,18]. To be consistent with our previous reports concerning ZP proteins [19,20], this paper uses the Spargo and Hope nomenclature system. However, other reports referenced in this paper use the numerical nomenclature system for avian ZP proteins, where ZPA = ZP2, ZPB1 = ZP1, ZPB2 = ZP4 and ZPC = ZP3.

These ZP proteins exhibit distinct expression patterns during follicular development. Avian ZPB1 is synthesized by the liver and subsequently transported to the developing follicles [15,20,21]. In contrast, ZPC and ZPD are both synthesized by the granulosa cells of the large preovulatory follicles [14,16,22,23]. Recent studies have revealed that the mRNA and protein of ZPA and ZPB2 are expressed in immature oocytes of the prehierarchical white follicles, and the expression levels gradually decrease as the follicles mature and increase in size [18,19,24,25,26]. Interestingly, ZPA protein expression was reported to only be detected in the GD region of the IPVL in mature follicles, likely because of this early expression of ZPA in the immature white follicles [25]. Similarly, the expression of ZPB2 mRNA is greater in immature white follicles, and the expression of ZPB2 mRNA was reported to be significantly greater in the GD granulosa cells when compared with the NGD granulosa cells in F_1_ and F_2_ follicles in turkey hens [19], suggesting that it may also be a factor in the preferential binding of sperm to this region of the IPVL. Additional research suggests that avian ZPB1 and ZPC may be involved in sperm binding to the IPVL or the initiation of the acrosome reaction [27,28].

The Ohio Agriculture Research and Development Center maintained several genetic lines of turkey hens; two of these lines of hens differ in growth rate, egg production and fertility [29,30,31]. Hepatic mRNA expression for ZPB1 was greater in the genetic E-line, which has improved fertility [20]. The mRNA expression of ZPC for the four largest follicles was greater in the genetic line with lower fertility, the F-line, and significantly increased in the NGD granulosa cells, as compared with the GD granulosa cells, in both genetic lines of turkey [20]. In contrast to ZPC, ZPB2 mRNA had higher expression in the GD region granulosa cells as compared with the NGD region, in both genetic lines of turkey [19]. Previous research indicates that there can generally be a low correlation between mRNA expression and protein concentrations [32,33]; therefore, the differential expression at the protein level of these IPVL components, ZPB1, ZPB2 and ZPC, was investigated in the current research. Quantitative Western blot analysis classically uses a housekeeping protein, such as β-actin, to correct for protein loading and transfer efficiency, but because the IPVL is an extracellular layer composed mainly of ZP glycoproteins, there is not a suitable housekeeping protein. Therefore, Stain-Free technology, which negates the need for a housekeeping protein [34], allowed for the first quantitative analyses of the different avian ZP proteins in the extracellular IPVL.

## 2. Materials and Methods

### 2.1. Animals

Two lines of turkey hens that differ in body weight (BW) and egg production were used in the present experiment. The E-line was selected from an established randomly bred control line [35,36], whereas the F-line was selected for increased 16 weeks BW [37]. The E line was originally selected based on the total number of eggs produced by dams for 84 d (generations 1 to 3). Selection was subsequently based on 180 d egg production for generations 4 to 26 and for 250 d egg production for subsequent generations [30]. Selection for increased BW in the F-line resulted in reduced egg production due to a decrease in the intensity of egg laying, as measured by average clutch size [38,39]. In contrast, selection for increased egg production in the E-line has reduced broodiness while increasing the intensity of egg laying [29]. Eggs produced by the F-line hens also have reduced rates of sperm penetration of the IPVL and fertility compared with eggs produced by the E-line hens [31,40].

Poults from the E-line and F-line were hatched at the Ohio Agriculture Research and Development Center and were soon shipped to the North Carolina State University turkey educational unit. The turkeys were raised in floor pens and provided 10 h of light per d until 25 weeks of age, when the h of light per d was reduced to 8. At 31 weeks of age, the turkey hens were moved to breeding pens (6 birds per pen) and photostimulated for reproduction by providing them 14 h of light per d. Each breeding pen was equipped with a nest box. The turkeys were always provided with free access to appropriate commercial diets and water through rearing and production. All turkey hen experimental animal procedures were approved by the North Carolina State University Animal Care and Use Committee.

### 2.2. Tissue Collection

To determine if there were differences in the protein expressions of ZPB1, ZPB2 and ZPC, unfertilized eggs produced from the hens from each genetic line were collected over a two-day period when the hens were 41, 43 and 44 weeks of age. Each egg was broken and the yolk was manually separated from the albumen. The yolk was placed into Krebs-Ringer bicarbonate buffer (pH 7.4) and a 1 cm^2^ section of the IPVL around the GD and a NGD area, on the opposite side of the follicle to the GD area, was collected. To obtain enough protein for subsequent Western blot analyses, the IPVL samples from the GD and NGD regions were pooled from three eggs from different hens of the same genetic line for each sample. Six samples per genetic line were collected, frozen, and stored at −80 °C in 150 μL of lysis buffer containing protease inhibitors [41]. The pooled IPVL cell lysates from the GD and NGD regions of each line were centrifuged at 20,000× *g* for 30 min at 4 °C. The supernatant fraction was recovered, and sample protein concentration was determined using the Bio-Rad Quick Start^TM^ Bradford protein assay (BioRad, Hercules, CA, USA) with bovine serum albumen (BSA) used as the standard.

### 2.3. Antibodies

Peptide fragments were selected for ZPB1, ZPB2 and ZPC which were 10–21 amino acids in length and did not contain potential glycosylation sites, phosphorylation sites or many hydrophobic amino acids. Additionally, the peptide fragments for each ZP protein were selected to be outside of the common ZP domain [41] and had 100% sequence identity between chicken and turkey for each ZP protein of interest. The peptide fragments selected, following consultation with Bio-Synthesis, Inc. (Lewisville, TX, USA), were subjected to BLAST homology analysis at http://www.ncbi.nlm.nih.gov/ (accessed 3 June 2004) to ensure the exclusion of unwanted cross-reaction with other proteins. The peptide sequence chosen for ZPB1 was PAGYEILRDEKVHGHQRPDRG-amidated, which corresponds to amino acids 127–147 of the mature protein. The peptide sequence selected for ZPB2 was ATPSINPHQQTQWPVLVNG-amidated, which corresponds to amino acids 378–396 of the mature protein. The peptide sequence picked for ZPC was acetylated- SWGAEAHSRAVAGSHPVAVQC, which corresponds to amino acids 47–67 of the mature protein. The chosen ZPB1, ZPB2 and ZPC peptide fragments were synthesized by Bio- Synthesis, Inc. (Lewisville, TX, USA) to a purity of >90%.

A portion of each of the synthesized protein fragments was then conjugated to keyhole limpet hemocyanin (KLH) and then used by Bio-Synthesis, Inc. (Lewisville, TX, USA), to produce rabbit polyclonal antibodies. Conjugated synthetic ZP peptides were injected into rabbits with Freund’s complete adjuvant at week 0, and Freund’s incomplete adjuvant at weeks 2, 4, 6 and 8. Two New Zealand White female rabbits at 12 weeks of age were immunized for each ZP peptide (six rabbits total). Serum was collected at 0-, 6-, 8- and 10-weeks post-injection and following exsanguination at 12 weeks of age.

Validation of synthetic antibodies was confirmed with Western blot analysis of each synthetic peptide using either preimmune serum obtained from a rabbit prior to inoculation with chicken synthetic peptide ZPB1, ZPB2 or ZPC or using serum obtained 10 weeks post-inoculation. SDS-PAGE gels containing 40 μg of turkey IPVL samples were transferred to a PVDF membrane (Immobilon-P), as previously described [42]. To determine the kilodalton (kDa) size of turkey ZPB1, ZPB2 and ZPC protein, the Spectra Multicolor Broad Range Protein Ladder (Fermentas, Glen Bernie, MD, USA) was electrophoresed in one lane of each gel. After blocking for 1 h in Tris-buffered saline, pH 7.6, containing 5% non-fat dried milk (LabScientific, Inc., Livingston, NJ, USA), the protocol and reagents for the Amersham ECL (enhanced chemiluminescence) Plus Kit (Amersham Biosciences, Piscataway, NJ, USA), along with goat anti-rabbit IgG-HRP conjugate (BioRad, Hercules, CA, USA), were utilized to detect ZPB1, ZPB2 and ZPC proteins using the antibodies produced against the synthetic peptides for these proteins. Chemiluminescence was detected using Super RX Fuji Medical X-Ray Film. Films were pan-developed using Kodak GBX developer and fixer/replenisher. Bands of 95 kDa, 59 kDa and 42 kDa were detected following Western blot analysis of isolated IPVL with anti-ZPB1, anti-ZPB2 and anti-ZPC, respectively (Figure 1).

### 2.4. SDS-PAGE and Western Blot Analysis

For each sample, 50 µg of IPVL protein lysate was mixed 1:1 with standard sample buffer containing 8 M urea, 2 M thiourea, 3% (wt/vol) SDS, 75 mM _DL_-dithiothreitol and 25 mM TrisHCl at pH 6.8, heated at 95 °C for 5 m, cooled and loaded into the gel. Fifty micrograms of protein was loaded into each lane of Mini-PROTEAN TGX Stain-Free 10% precast gels (Bio-Rad) alongside Precision Plus Protein All Blue Prestained Protein Standards (Bio-Rad). Gels were subjected to SDS-PAGE in 1X Tris-glycine-SDS (25 mM Tris, 192 mM glycine, and 0.1% (*w*/*v*) SDS, pH 8.3) running buffer under 70 V for 10 m followed by 120 V for 60 min in a Mini-PROTEAN Tetra Cell system (Bio-Rad). After SDS-PAGE, gels were UV-activated using a ChemiDoc MP Imaging System (Bio-Rad). Proteins were then transferred from the activated gels to Immun-Blot polyvinylidene difluoride membranes (Bio-Rad) in Towbin transfer buffer (25 mM Tris base, 192 mM glycine, and 20% (*v*/*v*) methanol) using a wet-blot transfer system (Bio-Rad) at 100 V for 1 h. The polyvinylidene difluoride membranes were washed briefly in Tris-buffered saline (0.02 M Tris base and 0.15 M NaCl, pH 7.4) containing 0.1% Tween 20 (TBST), followed by blocking in 5% BSA (Sigma-Aldrich, St. Louis, MO, USA) in TBST for 30 m. The membranes were then imaged using the ChemiDoc MP Imaging System (BioRad) for total protein normalization to determine the relative protein abundance.

The rabbit serum antibodies were purified using the Dynabeads Protein A Immunoprecipitation kit (Life Technologies, Carlsbad, CA, USA). The purified primary ZPB1, ZPB2 and ZPC antibodies were incubated at 1:1000 dilutions in TBST (w/2.5% BSA) with gentle rocking overnight at 4 °C. Following a washing procedure (3 times for 10 min each in TBST), goat anti-rabbit IgG-HRP conjugate (Bio-Rad) was incubated at a 1:10,000 dilution in TBST (w/2.5% BSA) with gentle rocking for 1 h at 20 °C. After a second washing procedure in TBST, Clarity Max Western ECL substrate mixture (Bio-Rad) was used for chemiluminescence detection using the ChemiDoc MP Imaging System (Bio-Rad), according to manufacturer’s protocol. The relative expression levels of ZPB1, ZPB2 and ZPC were then obtained by normalizing the density of the protein bands to the corresponding Stain-Free blot image (total protein normalization) in Image Lab software (Version 5.2.1, Bio-Rad). For each replicate blot, the quantification values of ZPB1, ZBP2 or ZPC were normalized for minimal loading differences in Image Lab software; the sample with the highest expression was assigned a value of 1 and the remaining samples were given expression quantification values relative to this sample with the most expression. Therefore, all replicate data for ZPB1, ZPB2 and ZPC are expressed as the fold-difference relative to the sample with the highest expression.

### 2.5. Statistical Analysis

In each experiment, the data were subjected to ANOVA according to the General Linear Model (GLM). Tukey’s multiple comparison procedure [42] was used to detect significant differences among genetic lines and IPVL locations. Differences were considered significant when *p* < 0.05. All statistical procedures were completed with Minitab Statistical software package (Release17, State College, PA, USA).

## 3. Results

### 3.1. Antibodies

The avian ZP antibodies produced by rabbits against a synthetic peptide segment of ZPB1, ZPB2 and ZPC were able to detect specific proteins in IPVL membranes isolated from laid turkey eggs. The antibody for ZPB2 reacted well with a 59 kDa band, which was not detected with preimmune serum (Figure 1A); this band was used for quantitative Western analysis of ZPB2. This molecular weight of 59 kDa closely agrees with a molecular weight of 58 kDa reported for quail ZPB2 [18]. Western blot analysis for ZPB1 on protein isolated from turkey egg IPVL detected bands of 95 kDa and 55 kDa that were not detected with preimmune serum (Figure 1B). Only the 95 kDa band was used for analysis of relative ZBP1 protein expression in this study because it agrees with the previously reported molecular weight for chicken ZPB1 [15]. The immune serum produced against the ZPC peptide fragment was tested in turkey hen protein samples extracted from the IPVL obtained from laid turkey eggs. The antibody for ZPC reacted well with a 42 kDA band, which was not detected with preimmune serum (Figure 1C); 42 kDa agrees with the previously reported molecular weight of chicken ZPC [14,22]. The 42 kDa band was used for quantitative Western analysis of ZPC.

### 3.2. ZPB2

Quantitative Western blot analysis showed that the relative expression of ZPB2 in IPVL from the GD region (0.957 ± 0.048) of laid turkey eggs was significantly greater than the NGD region (0.755 ± 0.068) (Figure 2A). However, there was no difference in relative ZPB2 signal intensity between IPVL isolated from E-line (0.581 ± 0.067) and F-line (0.613 ± 0.076) hens (Figure 2B).

### 3.3. ZPC

The relative abundance of ZPC in the NGD region (0.477 ± 0.109) tended to be higher (*p* = 0.09) than the GD region (0.237 ± 0.109) of the IPVL (Figure 3A). However, there was a significantly greater abundance of ZPC in IPVL isolated from E-line (0.500 ± 0.114) eggs relative to F-line (0.213 ± 0.072) eggs (Figure 3B). Comparison of relative abundance of ZPC based on location of the IPVL within the genetic lines revealed a higher relative abundance of ZPC in the NGD region of E-line hens and lower relative expression of ZPC in the GD region of F-line hens (Figure 4).

### 3.4. ZPB1

Quantitative Western blot analysis indicated that the relative expression of ZPB1 bands in IPVL from the NGD region (0.897 ± 0.066) of laid turkey eggs was significantly greater than the GD region (0.581 ± 0.086, Figure 5A). Additionally, there was significantly greater abundance of ZB1 in IPVL isolated from F-line (0.578 ± 0.093) eggs relative to E-line (0.293 ± 0.048) eggs (Figure 5B). Comparison of relative abundance of ZPB1 based on location of the IPVL within the genetic lines revealed relatively lower ZPB1 abundance in the GD region of E-line hens (0.220 ± 0.048) compared with the IPVL from the NGD region of F-line hens (0.721 ± 0.123, Figure 6).

## 4. Discussion

This study is the first report concerning the relative abundance of avian ZP proteins in the IPVL of oviposited eggs. The use of Stain-Free technology allowed for protein normalization [34] and quantitative Western analysis of this extracellular protein layer around the ovulated ovum. Additionally, the current study builds upon previous research that has reported differences in the mRNA expression of avian ZP proteins in E-line and F-line hens, as well as in the GD and NGD regions [19,20].

The abundance of ZPB2 protein was greater in the GD region of the IPLV, compared with the NGD region, of laid eggs in the two genetic line of turkey hens. This result agrees with previous mRNA research on these two genetic lines [19], as granulosa cells from the GD region of the F_1_ and F_2_ follicles in both genetic lines had a greater expression of ZPB2 mRNA than NGD granulosa cells. Avian sperm preferentially bind to the GD region of the ovulated ovum [7,9,10,12]; thus, the current research provides further evidence supporting the hypothesis that the greater abundance of ZBP2 protein at the GD region of the IPVL, relative to the NGD region, accounts for the increased number of sperm penetrations at the GD region. Additional research will be necessary to determine whether ZPB2 acts as a sole sperm receptor or if it works in concert with other ZP proteins such as ZPA, ZPB1, ZPC or ZPD to form the requisite sperm receptor complex. Although ZPB1, ZPC and ZPD have been reported to bind sperm in one or more avian species [16,27,28], none of these ZP proteins have a greater expression in the GD region relative to the NGD region. Only ZPB2 in the current study, and ZPA, as previously reported [25], are associated with higher expression in the GD region.

The avian sperm receptor is likely a 3D complex consisting of multiple entities, possibly in the form of other ZP proteins, which have previously been suggested to play roles in sperm binding. Previous reports indicate that both ZPA and ZPB2 are more highly expressed in the prehierarchical follicles [19,25]. It seems reasonable to suggest that when ZPA and ZPB2 are expressed early in development and localized at the GD region, both may contribute to an anchor network to which other ZP components (ZPB1 ZPC and/or ZPD) may bind when they are subsequently added to the IPVL during follicular maturation in hierarchical follicles. Upon the binding of one or more of these later produced ZP protein complexes to the ZPB2/ZPA network at the GD, specific 3D ZP protein motifs found almost exclusively at the GD region may form and serve as the sperm receptor. Additional research will have to be conducted that observes the structural biology of ZP proteins at the GD region to determine the exact confirmation, as well as knockout studies that facilitate a better understanding of the function or loss of function of ZP proteins such as ZPB2 and ZPA.

Hepatically expressed avian ZPB1, which is regarded as closely related to ZPB2 based on structural conservation [24], has also been suggested to play a role in initiating sperm binding to the IPVL [26]. Previously, it was reported that the hepatic mRNA expression of ZPB1 was greater in E-line turkey hens, selected for egg production and having higher fertility, than in the F-line hens, selected for their rapid growth [19,20]. However, due to the hepatic expression of ZPB1 mRNA, the relative expression based on IPVL location (GD vs. NGD) has not been investigated. The expectation in the current research was that the abundance of ZPB1 protein in the IPVL of eggs produced by hens from these genetic lines would reflect the hepatic mRNA expression data. However, it was the IPVL of the eggs from the F-line hens that exhibited a greater abundance of ZPB1 in the IPVL. Within the IPLV, the abundance of ZPB1 protein was greater in the NGD region than the GD region in turkey eggs.

This discrepancy between the hepatic mRNA expression and IPVL protein abundance of ZPB1 between the two genetic lines of turkey hens may relate to differences in follicular maturation amongst the two lines. F-line hens have an egg production rate that is less than half the rate of the E-line hen and have an extended hierarchy (F_1_–F_16_) relative to the E-line hen (F_1_–F_10_), but the hierarchical follicle size relative to BW is smaller in F-line hens [43]. The hepatic expression of avian ZPB1 is well established, although the mechanism involved in its transport to the developing follicles is not. Previous research indicates that ZPB1 protein expression is only detectable in follicles containing yellow yolk [27] and ZPB1 expression is highly responsive to estrogenic compounds [15,21]. Estrogen stimulates the hepatic expression of both vitellogenin and specialized VLDL particles that make up the yellow yolk; therefore, it may be that ZPB1 is transported along with these components or in a similar independent receptor-mediated process to the developing follicles. F-line hens have an extended hierarchy, lower rate of egg production, and produce smaller follicles relative to body size when compared with E-line hens [44]. This would allow for a longer period of ZPB1 deposition in the IPVL of a relatively smaller follicle and could contribute to the greater overall concentration of ZPB1 in the IPVL of F-line hens.

Structurally, avian ZPB1 has a repeated sequence region that consists of a tandem repeat of 20 proline-rich segments of about 24 amino acid residues [15]. Proline residues are rich in some extracellular protein forming fibrous aggregates, such as collagen and elastin, and there is a relationship of high proline content with elastomeric fibril formation on some natural elastic proteins and model peptides [45]. The liver-specific synthesis of avian ZPB1 might contribute to the rapid expansion of the IPVL throughout the rapid growth phase in hierarchical follicles and the proline-rich repeated sequence regions of ZPB1 may play a role in providing an elastic nature to the IPVL matrix during the rapid growth of hierarchical follicles [43]. This needs to be investigated in future research.

The abundance of ZPC protein in the IPLV from E- and F-line hens did not directly correlate with previously reported granulosa cell expression [20]. The expression of ZPC mRNA, from the four largest hierarchical follicles, was greater than in the F-line hens compared with the E-line hens [20]. However, the relative abundance of ZPC protein was greater in the E-line hens compared with the F-line hens, following quantitative Western analysis. The mRNA expression for ZPC was greater in the NGD than the GD granulosa cells for the F_1_ follicle of the F-line hens [20]. Although, in the current study, the abundance of ZPC in the NGD IPVL compared with the GD IPVL was numerically greater, it was not statistically significant (*p* = 0.07). A previous study that analyzed differences in the transcriptome between GD vs. NGD granulosa cells also reported an increase in ZPC from NGD granulosa cells [5]. The increased abundance of ZPC in the E-line compared with the F-line is directly in contrast to to the situation in the F-line, where ZPB1 abundance was greater in the F-line compared with the E-line. Given the greater fertility and number of sperm penetrations at the GD region of the E-line hens versus the F-line hens, these differences in ZPC and ZPB1 protein expression and fertility need further investigation, especially considering ZPB2 expression being equal between the two lines.

Previous research has indicated that our understanding of how mRNA and protein expression levels are related necessitates re-evaluation. Studies in animals demonstrate that the correlation coefficient (R^2^) between mRNA and protein can range anywhere from 0.09 to 0.46 [32]. Thus, only 9–46% of protein expression levels could be directly explained by the mRNA expression levels alone. Therefore, at the very most, 56% of the protein level expression observed must be explained by other factors. In another study, the mRNA and sequence signature could only account for 67% of protein expression levels [33]. In the current study, the R^2^ between protein expression levels and our previously reported mRNA expression levels ranged from 0.98 for ZPB2 between GD and NGD regions to 0.00 for ZPB1 between E- and F- line hens. These observations suggest that mRNA expression levels should be used as a general correlation with protein expression levels and should not be used as a direct predictor for protein abundance. Considering that ZPD mRNA did not differ between GD and NGD region granulosa cells in either E- or F-line turkey hens [20], it would be interesting to follow up with quantitative Western analysis of ZPD in light of this correlation between ZP mRNA expression and ZP protein abundance in the IPVL.

## 5. Conclusions

There have been several reports investigating the sperm binding potential and expression of different avian ZP proteins; however, this is the first study to investigate the relative abundance of different ZP proteins in the avian IPVL. The current results reveal that there are relative differences in IPVL protein abundance based on either location on the IPVL (GD region or NGD region) or genetic lines known to differ in fertility (E-line, higher fertility and F-line, lower fertility). Interestingly, most of the results differed from previously reported mRNA levels between E-line and F-line hens. However, both mRNA and protein levels of ZPB2 were more highly expressed in the GD region. Thus, the current research strongly implicates that ZPB2 likely plays a critical role in the preferential binding of sperm at the GD region of the ovulated ovum. Furthermore, the increased abundance of ZPB1 in the F-line hens and increased ZPC abundance in the E-line hens may reflect structural contributions of these ZP proteins to the rapidly growing hierarchal follicles that contribute to differences in sperm binding and fertility in these lines. Ultimately, this research provides important insights into the protein makeup of the avian IPLV, and a greater understanding of this layer can provide targets, such as ZPB2, for genetic selection to improve fertility in commercial breeding stocks.

## Figures and Tables

**Figure 1 animals-12-01672-f001:**
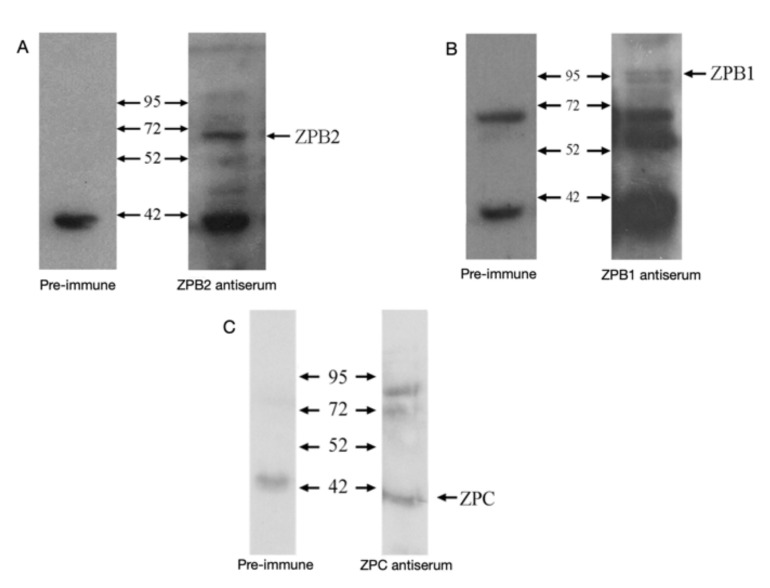
Western blot analyses for validation for anti-ZPB2 (**A**), anti-ZPB1 (**B**) and ZPC (**C**). The provided kDa sizes are based on the Spectra Multicolor BroadRange Protein Ladder that was electrophoresed and transferred from a gel that also contained duplicate 40 μg samples of total protein isolated from the GD region of the IPVL from laid turkey eggs. The PVDF membrane containing the samples was cut into two strips, each containing the two protein samples, and then analyzed with preimmune serum or with immune serum containing the respective chicken anti-ZP antibodies.

**Figure 2 animals-12-01672-f002:**
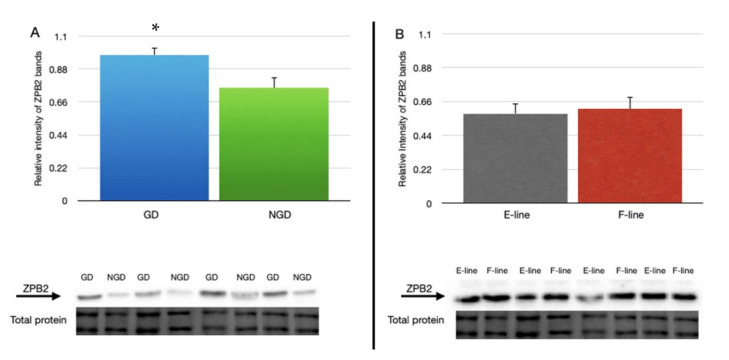
Relative abundance of ZPB2 protein in the perivitelline layer of laid eggs between (**A**) the non-germinal disc (NGD) region and germinal disc (GD) region with a representative blot for ZPB2 comparing GD and NGD regions and the section of total protein used for normalization. (**B**) Relative abundance of ZPB2 in the perivitelline layer of laid eggs between two genetic lines of turkey hens selected for either egg production (E-line) or rapid body growth (F-line) along with a representative blot for ZPB2 and total protein. For (**A**), *n* = 12 replicate samples per perivitelline location (6 E-line eggs plus 6 F-line eggs), and for (**B**), *n* = 12 replicate samples (6 GD plus 6 NGD) per genetic line. Values represent the mean ± SEM. * indicates significant difference in means, *p* < 0.05.

**Figure 3 animals-12-01672-f003:**
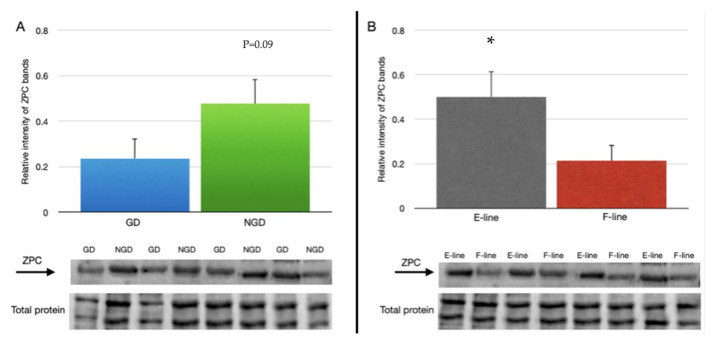
Relative abundance of ZPC protein in the perivitelline layer of laid eggs between (**A**) the non-germinal disc (NGD) region and germinal disc (GD) region with a representative blot for ZPC comparing GD and NGD regions and the section of total protein used for normalization. (**B**) Relative abundance of ZPC in the perivitelline layer of laid eggs between two genetic lines of turkey hens selected for either egg production (E-line) or rapid body growth (F-line) along with a representative blot for ZPC and total protein. For (**A**), *n* = 12 replicate samples per perivitelline location (6 E-line eggs plus 6 F-line eggs), and for (**B**), *n* = 12 replicate samples (6 GD plus 6 NGD) per genetic line. Values represent the mean ± SEM. * indicates significant difference in means, *p* < 0.05.

**Figure 4 animals-12-01672-f004:**
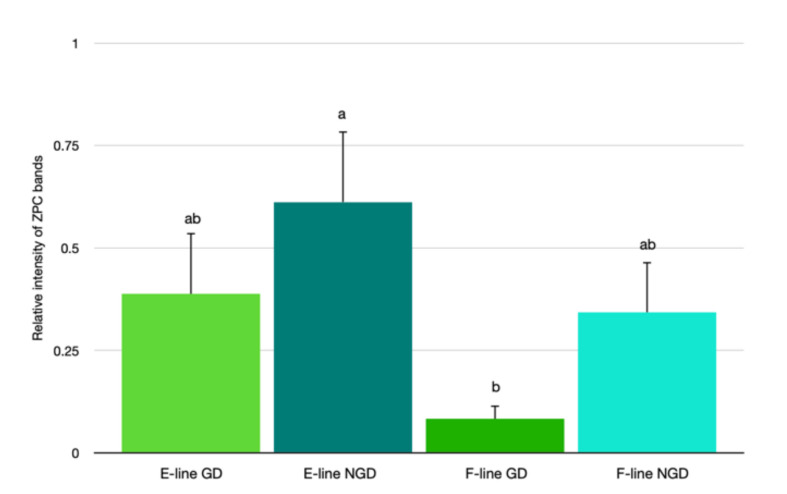
Relative abundance of ZPC protein in the germinal disc (GD) or the non-germinal disc (NGD) region of the perivitelline layer from two genetic lines of turkeys selected for egg production (E-line) or rapid body growth (F-line). Values represent the mean ± SEM. *n* = 6 replicates per sample. Means with different letters differ, *p* < 0.05.

**Figure 5 animals-12-01672-f005:**
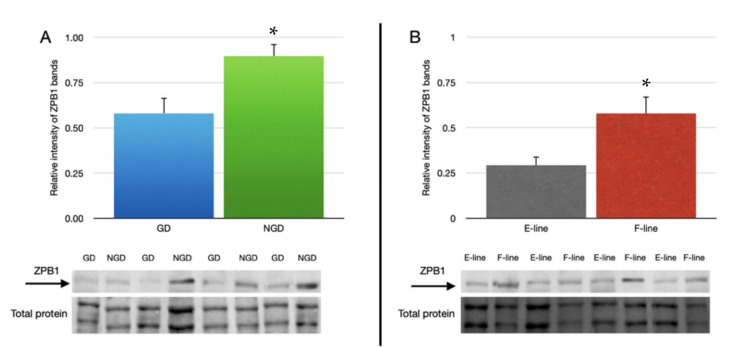
Relative abundance of ZPB1 protein in the perivitelline layer of laid eggs between (**A**) the non-germinal disc (NGD) region and germinal disc (GD) region with a representative blot for ZPB1 comparing GD and NGD regions and section of total protein used for normalization. (**B**) Relative abundance of ZPB1 in the perivitelline layer of laid eggs between two genetic lines of turkey hens selected for either egg production (E-line) or rapid body growth (F-line) along with a representative blot for ZPB1 and total protein. For (**A**), *n* = 12 replicate samples per perivitelline location (6 E-line eggs plus 6 F-line eggs), and for (**B**), *n* = 12 replicate samples (6 GD plus 6 NGD) per genetic line. Values represent the mean ± SEM. * indicates significant difference in means, *p* < 0.05.

**Figure 6 animals-12-01672-f006:**
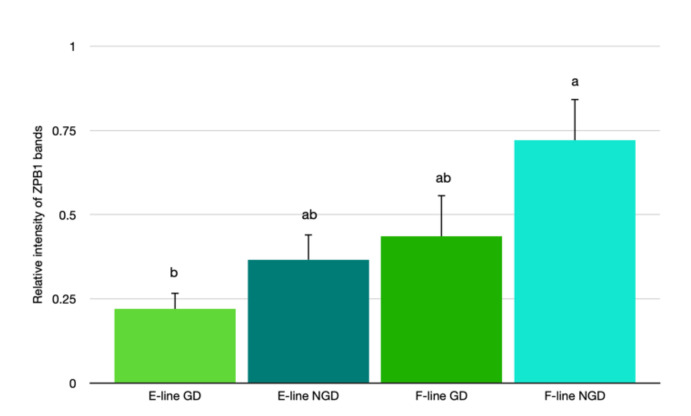
Relative abundance of ZPB1 protein in the germinal disc (GD) or the non-germinal disc (NGD) region of the perivitelline layer from two genetic lines of turkeys selected for egg production (E-line) or rapid body growth (F-line). Values represent the mean ± SEM. *n* = 6 replicate samples. Means with different letters differ, *p* < 0.05.

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
