# Peer review of "Quantitative Protein Analysis of ZPB2, ZPB1 and ZPC in the Germinal Disc and a Non-Germinal Disc Region of the Inner Perivitelline Layer in Two Genetic Lines of Turkey Hens That Differ in Fertility"

_animals, 2022, doi:10.3390/ani12131672_

Round 1
Reviewer 1 Report
Quantitative protein analysis of ZPB2, ZPB1 and ZPC in the germinal disc and a non-germinal disc region of the inner perivitelline layer in two genetic lines of turkey hens that differ in fertility.
by Andrew Benson, Josh Steed, Mia Malloy and Adam Davis
Nomenclature of ZP proteins used in text: ZPA, ZPB1, ZPB2, ZPC, ZPD
For quite some time now, a numerical nomenclature has been used for ZP proteins in mammals and in other vertebrates. The nomenclature used by the authors can therefore be confusing: I had to look up, e.g., what is ZPB1 or ZPB2. Also, in the references in the text - relevant to these proteins - the nomenclature varies between different authors.
For example,
ref. # 18 refers to ZP4 but here (3.1) it is "......weight of 58 kDa reported for quail ZPB2 [18]"
ref. # 15 refers to ZP1 but here (3.1) ".......... molecular weight for chicken ZPB1 [15] "
ref. #14/20 refers to ZP3/ZPC but here (3.1) "....... molecular weight of chicken ZPC [14, 20] "
This can be confusing for the reader, also when looking up molecular weights of these proteins.
Suggestion: include a note which gives the "translation", e.g. ZPA = ZP2, ZPB1 = ZP1, ZPB2 = ZP4, ZPC =ZP3.
typos:
3.3 should be 3.4
Fig. 6 is Fig. 5 in text
Fig. 7 is Fig. 6 in text
Results
The results of the paper are based on ZP protein quantification with specific anti-ZP synthetic peptide antibodies, detected on Western immunoblots.
ZP1 and ZP4 share a ZP domain with high similarity and it is quite possible that the anti-ZP4 synthetic peptide antibody (made against sequence 378-396, ZP domain) crossreacts with with ZP domain of ZP1 (high similarity).
Although relative abundance of ZP1, ZP4 or ZP3 was estimated according to molecular weight - ZP1 has a higher Mr than ZP4 – the blots shown in Fig. 1 (B) and (C) are not very convincing. Is ZP1 really ZP1, is ZP3 really ZP3 ?
Sequencing the bands determined to be ZP1, ZP4 and ZP3 should confirm the data, and would enhance the significance of the data.
General
- Interpretation of data is good.
- Interesting points about fertility in E-line/F-lines in turkey hens, data on differences in protein composition in IPVL depending on the regions from where IPVL was collected, and differences in protein abundance compared with respective mRNA expression.
- Discussion about structural implications of ZP protein expression and location and possible sperm-binding region/sperm-binding ZP protein is interesting.
Author Response
I completely agree with the first critique and the confusion with the nomenclature. We used the Spargo and Hope nomenclature to keep consistency among our papers reporting on ZP proteins in turkey hens. The other labs reporting on avian ZP proteins have continued to use the numerical system of nomenclature for ZP proteins, which can make some of the references noted above confusing. We have added the following statement in the Introduction (lines 65-58) to address this helpful critique:
“To be consistent with our previous reports concerning ZP proteins [19-20], this paper will be using the Spargo and Hope nomenclature system. However, other reports referenced in this paper use the numerical nomenclature system for avian ZP proteins where ZPA = ZP2, ZPB1 = ZP1, ZPB2 = ZP4 and ZPC = ZP3.”
We have also corrected the noted typos below that were present in the submitted manuscript. Thank you for pointing them out:
3.3 should be 3.4. Corrected
Fig. 6 is Fig. 5 in text. Corrected
Fig. 7 is Fig. 6 in text. Corrected
For the critique concerning the possible cross-reaction between ZPB1 and ZPB2, I added a clarifying statement in the Materials and Methods section under 2.3 Antibodies (lines 143-145):
"Additionally, the peptide fragments for each ZP protein were selected to be outside of the common ZP domain [43], and have 100% sequence identity between chicken and turkey for each ZP protein of interest."
Since all ZP proteins share a common ZP domain, we were careful in picking an antigenic sequence that was outside of the common ZP domain but highly conserved among both Turkey and Chicken. Given that anit-ZPB1 detected a 95 kDa (as predicted) and anti-ZPB2 detected a 59 kDa band (as expected), we can assure that analysis of these respective bands are associated with ZPB1 and ZPB2 expression. Furthermore, the antigenic sequence for ZPB1 (PAGYEILRDEKVHGHQRPDRG) has no overlap with the antigenic sequence for ZB2 (ATPSINPHQQTQWPVLVNG).
Concerning the critique for Figure 1:
The basis of using the 42 kDa band for ZPC and the 95 kDa band for ZPB1 was based on both reported molecular weights for the two avian ZP proteins as well as the detection of these two bands with the antiserum and absence of these bands with the Pre-immune (as shown with Figure 1).
We also agree that sequence confirmation would add additional assurance, but we are unfortunately not in the position to complete that step at this point.
We appreciate your helpful critiques as we believe they have improved the paper. Thank you.
Reviewer 2 Report
Manuscript dealing with quantitative protein analysis of ZPB2, ZPB1, and ZPC in the germinal disc and a non-germinal disc region of the inner perivitelline layer in two genetic lines of turkey hens that differ in fertility. The abstract was appropriate. The introduction provides a suitable justification for this research. Materials and Methods contained detailed information. Results were well presented. The discussion was good overall and contained comparisons with plenty of relevant studies. Comparisons could have been presented in a more concise fashion. Appropriate conclusions.
Author Response
Thank you for your insight and review. We have taken into consideration the conciseness of the report and hope you are satisfied with the new submission.
Reviewer 3 Report
The authors analyzed relative abundances of avian ZP glycoproteins ZPB2, ZPC and ZPB1 (ZP4, ZP3 and ZP1, respectively, in the another nomenclature system) in both the germinal disc (GD) and nongerminal disc (NGD) regions of inner perivitelline layer (IPVL), the avian homolog of mammalian zona pellucida, in mature ovarian follicles of two genetic lines of turkey hens that are named E- and F-lines. In this study, the results suggested that the relative abundance of ZPB2 was significantly higher in the GD than in the NGD region of mature IPVL similarly to ZPA as suggested in their previous study. On the other hand, the results suggested that the relative abundances of ZPC tended to be higher and that of ZPB1 was significantly higher in the NGD in the GD region of mature IPVL. Furthermore, the results suggested that the relative abundance of ZP3 and ZPB1 were both significantly higher in the E-lines (egg production) and in the F-lines (rapid body growth), respectively.
Based on these and previous results, the authors discussed relationships of ZP-glycoprotein compositions in IPVLs with their physiological functions including sperm binding capacities of IPVLs. Although there might be synergy effects among sperm binding capacity of ZP glycoproteins and further studies are required to reveal three dimensional network of ZP glycoproteins, I think it is quite important to investigate the relative abundance of the components of multifunctional extracellular matrix such as IPVL.
I think this manuscript is suitable for publication if the authors consider my major and minor comments below.
Major comments:
1) In previous studies, the relative abundance of ZPA was higher in the GD than in the NGD region of mature IPVL. I recommend the authors to mention whether the relative amount of ZPA was.
2) I recommend the authors to mention the relative abundance of ZPD in IPVLs.
3) In a legend of Figure.1, the word "Autoradiograms" are used, although I could not find the procedure of Autoradiograph in the Materials and Methods section.
Minor comments
1) In line 186, there might be a typo (gentile –> gentle).
2) In line 363, there might bet a typo (is –>in)
Author Response
1. ZPA:
We completely agree including ZPA in the study would be interesting and strengthen the insight concerning the avian IPVL; However, we are unable to test differences since we did not make, nor have access, to an antibody for ZPA. The report concerning (Nishio et al., 2014) came out after we began the process of validating custom antibodies for proteins of interest. Also, this was a targeted follow-up to our previous research concerning ZPB1, ZPB2, and ZPC expression. As a result of this critique and because I also agree research concerning the relative abundance of ZPA is important, I added some more information concerning future studies looking at ZPA in the discussion section.
2. ZPD:
Unfortunately, we cannot provide this information since we do not currently have antibodies for ZPD. Our original hypothesis was to follow up on our previously reported differences in ZP mRNA expression between GD and NGD regions of the granulosa cells. We did not find any differences in ZPD expression between GD and NGD granulosa cells (nor E and F line hens). Thus, we aimed to target the proteins (ZPC and ZPB2) that we previously reported to be different between GD and NGD. We also thought ZPB1 would be an interesting target due to its difference between E and F line hens and since its relative abundance between GD and NGD regions has not been investigated. However, you do make a great point since our ZP protein data didn’t correlate with our previously reported levels of mRNA for both ZPC and ZPB1. Therefore, I added a statement in the discussion that future research should investigate the protein abundance of ZPD protein in light of the poor correlation of mRNA for both ZPB1 and ZPC.
3. Autoradiograms:
I truly appreciate you catching this mistake. “Autoradiograms” was removed as a description of the blots since it can be misleading as a description of how the Western blots were developed (we used chemiluminescence and not radiation for signal detection). Nevertheless, we added further detail concerning the validation Western blotting technique in the Materials and Methods section.
Minor Comments: Thank you for catching these mistakes:
1) In line 186, there might be a typo (gentile –> gentle). Corrected
2) In line 363, there might bet a typo (is –>in) Corrected
Thank you for your critiques and review of this paper. I certainly agree the addition of ZPA and ZPD data would provide more insight into the avian IPVL however we are not able to address this critique since we only made antibodies for those ZP proteins that had interesting mRNA differences in these turkey hens. We hope that our changes to the discussion section communicate our awareness and eagerness to pursue ZPA and ZPD protein research in future studies.
Round 2
Reviewer 3 Report
I think the revised version of this manuscript is suitable for publication.